Molecular cytogenetic analyses of Epinephelus bruneus and Epinephelus moara (Perciformes, Epinephelidae)

Guo Minglan 1 2
Wang Shifeng 3
Su Yongquan 2
Zhou Yongcan 3
Liu Min 2
Wang Jun 2 junw@xmu.edu.cn
1 Key Laboratory of Tropical Marine Bio-resources and Ecology, South China Sea Institute of Oceanology, Chinese Academy of Sciences , Guangzhou, Guangdong , PR China
2 College of Ocean and Earth Sciences, Xiamen University , Xiamen, Fujian , PR China
3 Hainan Key Laboratory for Sustainable Utilization of Tropical Bioresource, Hainan University , Haikou, Hainan , PR China
Song Linsheng
Electronic publication date: 2014 Jun 10
Publication date: 2014
Volume: 2
Electronic Location ID: e412
Received 2014 Apr 2; Accepted 2014 May 13
Copyright: © 2014 Guo et al.
Copyright year: 2014
Copyright holder: Guo et al.
License: This is an open access article distributed under the terms of the Creative Commons Attribution License, which permits unrestricted use, distribution, reproduction and adaptation in any medium and for any purpose provided that it is properly attributed. For attribution, the original author(s), title, publication source (PeerJ) and either DOI or URL of the article must be cited.
License URL: https://creativecommons.org/licenses/by/4.0/

Keywords: Species-specific, Fish, Cytogenetics, Chromosomes, Taxonomy, Evolution

Funding: Natural Science Foundation of China 40576064 31001124 31260644 31060355 The National Marine Public Welfare Research Project of China 201205025 This work was supported by the grants from the Natural Science Foundation of China (40576064, 31001124 and 31060355) and the National Marine Public Welfare Research Project of China (No. 201205025). The funders had no role in study design, data collection and analysis, decision to publish, or preparation of the manuscript.

==============================
Genus Epinephelus (Perciformes, Epinephelidae), commonly known as groupers, are usually difficult in species identification for the lack and/or change of morphological specialization. In this study, molecular cytogenetic analyses were firstly performed to identify the closely related species Epinephelus bruneus and E. moara in this genus. The species-specific differences of both fish species showed in karyotype, chromosomal distribution of nucleolar organizer regions (NORs) and localization of 18S rDNA. The heterochromatin (interstitial C-bands) and distribution pattern of telomere (TTAGGG)n in E. bruneus revealed the chromosomal rearrangements and different karyotypic evolutionary characteristics compared to those in E. moara. The cytogenetic data suggested that the lineages of E. bruneus and E. moara were recently derived within the genus Epinephelus, and E. moara exhibited more plesiomorphic features than E. bruneus. All results confirmed that E. moara, which has long been considered a synonym of E. bruneus, is a distinct species in the family Epinephelidae. In addition, molecular cytogenetic analyses are useful in species differentiation and phylogenetic reconstruction in groupers.

Introduction

The family Epinephelidae comprises approximately 163 grouper species in 16 genera (Craig, Sadovy de Mitcheson & Heemstra, 2011). These species are of considerable economic value, especially in the coastal fisheries of tropical and subtropical areas (Heemstra & Randall, 1993). Taxonomic confusion in the Epinephelidae often occurs due to similarities of color patterns and ontogenetic changes in color (Heemstra & Randall, 1993; Craig, Sadovy de Mitcheson & Heemstra, 2011). Epinephelus bruneus (Bloch 1793) and E. moara (Temminck and Schiegel 1842) are two important aquaculture and commercial fish species. However, E. moara has long been considered a synonym of E. bruneus due to their similarities in coloration and overlapping in geographical distributions (Heemstra & Randall, 1993; Craig, Sadovy de Mitcheson & Heemstra, 2011). Designation of correctly identified new species is important not only to the study of phylogenetic relationships, but also to the identification of fry and parent in grouper aquaculture. Based on morphological characteristics and molecular comparisons, E. moara has been suggested as a valid species (Guo et al., 2008; Guo et al., 2009; Liu et al., 2013). The interspecific differences between them were identified mainly based on the skeleton system as well as the meristic and morphometric characteristics (Guo et al., 2008). Gene differentiation (Guo et al., 2009) and mitogenome analyses (Liu et al., 2013) partially provided molecular information confirming their taxonomic status. Chromosomes are the carriers of genetic information, and chromosomal recombination plays a vital role in genetic diversity. Therefore, more other evidences are needed to support the hypothesis that E. moara is a valid species, such as molecular cytogenetic analyses.

Chromosomes are hereditary elements of the complete nuclear genome. Molecular cytogenetic studies on chromosomes constitute important approaches for characterizing species and reconstructing phylogenetic relationships (Galetti, Aguilar & Molina, 2000; Ocalewicz, Woznicki & Jankun, 2008; Cioffi, Martins & Bertollo, 2010; Ruiz-Herrera, Farre & Robinson, 2012). Karyological features indicate the evolutionary distance between species of different taxonomic categories (Dobigny et al., 2004). The nucleolar organizer regions (NORs) were particularly significant in chromosomal evolutionary analyses (Miller et al., 1976; Fujiwara et al., 1998). Heterochromatin corresponding to C-bandings is normally associated with rearrangements, quantitative variation, and formation of new karyotypes (Miklos & Gill, 1982; Rocco et al., 2002). Many taxonomic studies were based on the variations and polymorphism of the chromosomes containing major 18S rDNA (both active and non-active) (Cioffi, Martins & Bertollo, 2010; Britton-Davidian, Cazaux & Catalan, 2012), and minor 5S rDNA (Fujiwara et al., 1998; Mazzei et al., 2004) by fluorescent in situ hybridization (FISH). Location of telomeric sequence (TTAGGG)n provides direct evidence for cytotaxonomic studies and chromosomal evolution in fishes (Sola et al., 2003; Scacchetti et al., 2011). Therefore, molecular cytogenetic information has provided important contributions to the characterization of biodiversity and the evolution of ichthyofauna (Jesus et al., 2003; Vicari et al., 2008). The analyses of different methods above present a complete karyotypic picture for organisms.

The cytogenetic information provided by a variety of approaches will allow us to more fully explain the taxonomic and evolutionary statuses, and reveal the inherent differences of E. bruneus and E. moara. In this study, the karyotypic techniques, including Giemsa-staining, Ag-staining, C-banding and localization of 18S rDNA and telomere (TTAGGG)n by FISH, were used to investigate the molecular cytogenetic characteristics of E. bruneus and E. moara. Cytogentic data provided a better definition of the specific epithet for these cryptic species. Molecular cytogenetic analyses were found to be applicable in differentiating between closely related species and reconstructing phylogenetic relationships in groupers.

Materials and Methods

Fish collection and identification

Twenty-five individuals of E. bruneus (standard length, LS, 140–550 mm) and 24 individuals of E. moara (LS, 188–650 mm) were collected alive from the coastal waters of Fujian, China, and reared in laboratory for one week before analyses. Specimen identification was based on external coloration (Fig. 1), skeleton and morphological characteristics established in previous studies (Guo et al., 2008).

Figure 1 Specimen of adult E. bruneus and E. moara.

Specimen of adult E. bruneus (Bloch, 1973) [standard length (LS), 550 mm] and E. moara (Temminck & Schiegel, 1842) (LS, 650 mm) (Xiamen, Fujian, China, ML Guo).

Chromosome preparation, karyptyping and staining analyses

Fishes were injected with colchicine (3 µg/g weight, Sigma) for 30 min. Mitotic chromosomes were obtained from cell suspensions of anterior kidney after the fishes were anesthetized with tricaine methanesulfonate (MS222, 100 mg/L, Sigma), using the conventional air-drying method (Ojima, Hitotsumachi & Makino, 1966). Chromosomes were stained using Giemsa and classified as metacentric (m), submetacentric (sm), subtelocentric (st), or acrocentric (a) based on the arm ratios (Levan, Fredga & Sandberg, 1964). The nucleolar organizer regions (NORs) were visualized by Ag-staining (Howell & Black, 1980). Heterochromatin was identified by C-banding using barium hydroxide method (Sumner, 1972). After the acquisition of anterior kidney, tissue samples (mostly dorsal muscle) were collected and stored at −80 °C. All experiments were conducted in accordance with the guidelines approved by the Institutional Animal Care and Use Committee at Xiamen University.

Chromosomal probes preparation

Genomic DNA of all specimens was extracted from muscle tissue using the phenol-chloroform method (Sambrook, Fritsch & Maniatis, 1989). 18S rDNA and 5S rDNA probes for chromosome hybridization were prepared as follows: The partial coding region of 18S rDNA were amplified using the primers 18S rDNA-F (5′-GTAGTCATATGCTTGTCTC-3′) and 18S rDNA-R (5′-TCCGCAGGTTCACCTACGGA-3′) as described by White et al. (1990). The coding region of 5S rDNA were obtained using the primers 5S rDNA-F (5′-TACGCCCGATCTCGTCCGATC-3′) and 5S rDNA-R (5′-CAGGCTGGTATGGCCGTAAGC-3′) indicated by Martins & Galetti (1999). PCR reactions were performed as following: 94 °C for 4 min, followed by 35 cycles of 94 °C for 30 s, 54 °C (for 18S rDNA) or 62 °C (for 5S rDNA) for 1 min, and 72 °C for 1 min, and a final extension at 72 °C for 5 min. The nucleotide sequences of 18S rDNA and 5S rDNA were obtained after cloning into the pMD-18T vector (Takara, Japan), and subjected to Blastn in NCBI database (http://www.ncbi.nlm.nih.gov). Telomere probes for chromosome hybridization were prepared as follows: telomeric repeat sequences (TTAGGG)n were amplified by PCR using (5′-TTAGGG-3′)5 and (5′-CCCTAA-3′)5 as primers (Ijdo et al., 1991). All probes were labeled with biotin-16-dUTP (Roche, Germany) by nick translation according to the manufacturer’s instructions.

Fluorescence in situ hybridization (FISH)

FISH and probe detection were conducted using methods as described previously (Wang et al., 2010). Briefly, avidin-fluorescein isothiocyanate (FITC) (Sigma, USA) was used for signal detection of probes 18S rDNA, 5S rDNA and telomere (TTAGGG)n based on the manufacturer’s instruction. Chromosomes were counterstained with 1 µg/ml 4′, 6′-diamidino-2-phenylin-dole (DAPI) (Roche, USA) in anti-fade solution of 70% glycerol, 2.5% DABCO [1,4-Diazabicyclo (2.2.2) octan], and 1 × standard saline concentration (SSC) at pH 8.0. Hybridization signals were observed and analyzed under a fluorescence microscope Leica DM-400CCD.

Results

Karyotypes and banding patterns

A total of 172 metaphases of E. bruneus and 156 metaphases of E. moara were analyzed to determine the karyotype structure. All specimens of E. bruneus and E. moara invariably showed the same diploid number of chromosomes, 2n = 48. The karyotype formulas of E. bruneus and E. moara were 2m + 4sm + 42a, giving a fundamental number (NF) equaled to 54 (Figs. 2A and 2B), and 4sm + 44a, NF = 52 (Figs. 2C and 2D), respectively. Chromosomes pairs were numbered based on the relative length. The smallest chromosomes pairs No. 24 were submetacentric chromosomes (sm-3 for E. bruneus and sm-2 for E. moara, Figs. 2B and 2D). Chromosome pairs No. 9 in length were sm-2 for E. bruneus and sm-1 for E. moara. Chromosome pairs No. 2 were metacentric chromosomes m-1 for E. bruneus. Other chromosomes were acrocentric (a) chromosomes for both E. bruneus and E. moara.

Figure 2 Chromosome metaphase and corresponding karyotype of E. bruneus and E. moara.

Chromosome metaphase (left, Giemsa staining) and corresponding karyotype (right) of E. bruneus (A and B) and E. moara (C and D). Scale bar = 5 µm. a, acrocentric; m, metacentric; sm, submetacentric.

Active NORs were identified on the terminal position of short arms or sub-centromere regions of those biarmed chromosomes. In E. bruneus, five actively transcribed NORs were located on the metacentric and submetacentric chromosomes (Fig. 3A). In E. moara, four Ag-NORs were found on the submetacentric chromosomes (Fig. 3B).

Figure 3 Ag-NORs characteristics of E. bruneus and E. moara with silver staining.

Ag-NORs characteristics of E. bruneus (A) and E. moara (B) with silver staining. Thick black arrows indicate the chromosomes No. 9 in length, thin black arrows represent the chromosomes No. 24 in length, and hollow arrows show the chromosomes No. 1 in length. Scale bar = 5 µm.

The constitutive heterochromatin was observed in the centromeric and/or pericentromeric region of most chromosomes for both E. bruneus and E. moara. And the biarmed chromosome pairs with positive Ag-NORs were coinciding with the positive heterochromatin C-bandings. While three pairs of acrocentric chromosomes were almost indiscernible in both fish species (Figs. 4A and 4B). However, the significant differences of heterochromatin were the heterochromatic blocks found in the interstitial region of the long arms of one pair of medium-sized acrocentric chromosome in E. bruneus (Fig. 4A).

Figure 4 C-banding patterns of E. bruneus and E. moara.

C-banding patterns of E. bruneus (A) and E. moara (B). Heterochromatic blocks were observed in the interstitial region of the long arms of acrocentric chromosome pair No. 12 in E. bruneus (white hollow arrows). Heterochromatin C-bands was consistent with the positive Ag-NORs sites on chromosome pair No. 2 in both fish species (thick black arrows). Heterochromatin C-bands were indiscernible for three pairs of chromosomes (Red arrows). Thin black arrows represent the chromosomes No. 24 in length. Scale bar = 5 µm.

Sequences analyses

Sequences of 18S rDNA (GenBank accession nos. FJ176793 and FJ176794) and 5S rDNA (GenBank accession nos. FJ176796 and FJ176795) were amplified from genomic DNA of E. bruneus and E. moara. Sequence of 18S rDNA contained partial DNA of gene 18S rRNA. Partial DNA sequence of 5S rDNA included the encoding and non-transcribed spacer (NTS) region for both fish species. The determined sequences were highly conserved. The nucleotide similarities of partial 18S rDNA and 5S rDNA were 100% and 99.99%, respectively, for both fish species. The phylogenic neighbor-joining (NJ) trees based on partial sequences of 18S rDNA and 5S rDNA strongly support the closed relationship of E. moara and E. bruneus (high bootstrap values of 92 and 100). And genera of the order percomorpha were mostly reconstructed the phylogenetic relationship by partial sequences of 18S rDNA but not 5S rDNA (data not shown).

FISH analyses

Multiple sites of 18S rDNA by FISH confirmed the data obtained by Ag-staining for NORs. In E. bruneus, six positive signals (both active and non-active) were identified, corresponding to metacentric (m) and submetacentric (sm) chromosomes (Fig. 5A). Four hybridization signals were observed on the short arms of submetacentric chromosomes in E. moara (Fig. 5B). 5S rDNA and 18S rDNA were found on different chromosomes. Two 5S rDNA sites were located on the arms of a medium-sized acrocentric chromosome pair in both E. bruneus and E. moara (Figs. 6A and 6B).

Figure 5 Distribution of 18S rNDA on chromosomes of E. bruneus and E. moara by FISH.

Distribution of 18S rNDA on chromosomes of E. bruneus (A) and E. moara (B) by FISH. White arrows indicate the biarmed chromosome, pairs No. 2, No. 9 and No. 24 in length, in both fish species. Scale bar = 5 µm.

Figure 6 Localization of 5S rNDA on chromosomes of E. bruneus and E. moara by FISH.

Localization of 5S rNDA on chromosomes of E. bruneus (A) and E. moara (B) by FISH. White arrows indicate the two 5S rDNA clusters located on the arms of one of acrocentric chromosome pair. Scale bar = 5 µm.

Telomeric repeats of (TTAGGG)n showed the typically telomeric signals on both telomeres and/or centromeric region of all chromosomes in E. bruneus and E. moara. No positive signal was detected at interstitial sites (Figs. 7A and 7B). Ten chromosome pairs of E. bruneus were significantly stronger than the signals of the others (Fig. 7A). However, E. moara were characterized by uniform telomeric signals in strength and size (Fig. 7B).

Figure 7 Distribution of telomeric (TTAGGG)n sequence on chromosomes of E. bruneus and E. moara by FISH.

Distribution of telomeric (TTAGGG)n sequence on chromosomes of E. bruneus (A) and E. moara (B) with telomeric (TTAGGG)n sequence using FISH. Red arrows indicate chromosomes with significantly stronger and larger telomeric signals than others in E. bruneus. Scale bar = 5 µm.

Discussion

Our previous study has distinguished E.bruneus and E. moara as two species based on their morphometric and skeletal characteristics (Guo et al., 2008). The species-specific differences showed obviously on the bars of the body and stable skeleton characteristics of adult. And the pyloric caeca indicates their different feeding habits and digestive function, which means they could have different ecological niches. Mitogenome and molecular comparisons confirmed E. moara to be a valid species of the family Epinephelidae (Liu et al., 2013). Further, we developed a molecular method to differentiate both fish species (Guo et al., 2009). However, the cytogenetic backgrounds and evolutionary situation, which is very important to the cultivation and protection of fish resources, remains unclear for E. bruneus and E. moara. We here comprehensively analyzed the cytogenetic backgrounds, and reconstructed their phylogenetic relationships using molecular cytogenetic analyses.

For E. bruneus and E. moara, species-specific characteristics presented in karyotype, NORs, C-banding and telomere distribution patterns. Karyotype variation appears to parallel speciation events in many groups of vertebrates (Morescalchi et al., 2007; Ruiz-Herrera, Farre & Robinson, 2012). Variations of NOR constituted a strong cytotaxonomic character in fishes (Fujiwara et al., 1998; Galetti, Aguilar & Molina, 2000). Many species in genus Epinephelus showed the same karyotypic characteristics, such as karyotype formula and NORs (Wang et al., 2012). However, karyotype formula and NORs were different between E. bruneus and E. moara (Table 1). Further, interstitial C-bandings were observed in E. bruneus, but not in E. moara. Similar interstitial heterochromatin was also found in E. coioides (Wang et al., 2010) and Diplectrum radiale (de Aguilar & Galetti, 1997). The distribution patterns of (TTAGGG)n were different obviously between E. bruneus and E. moara. Cytogenetic differences were inter-specific, because E. bruneus and E. moara showed a similar geographical distribution (Heemstra & Randall, 1993; Guo et al., 2009), and coupled with heterogeneously morphological characteristics (Guo et al., 2008) and chromosomal structure.

Table 1 Available cytogenetic data of the genus Epinephelus.

Species	2n	Karyotype formula	FN	NORs	C-banding	Reference	
E. adscencionis	48	48a	48	SCR(24)a	C(1-24)	(Molina, Maia-Lima & Affonso, 2002)	
				TR(2)			
E. akaara	48	5st + 43a	48	/	/	(Wang et al., 2004)	
E. alexandrinus	48	48a	48	SCR(24)a	NC(1-23), SCR(24)a	(Martinez et al., 1989)	
E. awoara	48	48a	48	SCR(24)	NC(1-23), SCR(24)a	(Wang et al., 2012)	
E. bruneus	48	2m + 4sm + 42a	54	SCR(24, 9, 2)	NC (?)	Present study	
					C (?)		
					SCR(24,9,2)		
					SA(2)		
					IR (?)		
E. caninus	48	48a	48	SCR(24)	/	(Rodríguez-daga, Amores & Thode, 1993)	
E. coioides	48	2sm + 46a	50	EA(24)	C(1-11, 13-24)	(Wang et al., 2010)	
					SCR(5,12) EA(24)		
E. diacanthus	48	2sm + 46a	50	/	/	(Natarajan & Subrahmanyan, 1974)	
E. fario	48	4m + 6sm + 4st + 34a	62	/	/	(Zheng, Liu & Li, 2005)	
E. fasciatomaculosus	48	48a	48	SCR(24)	/	(Li & Peng, 1994)	
E. fasciatus	48	48a	48	SCR(24)	/	(Li & Peng, 1994)	
E. fuscoguttatus	48	2sm + 46a	50	/	/	(Liao et al., 2006)	
E. guaza	48	48a	48	SCR(24)a	NC(1-23) SCR(24)a	(Martinez et al., 1989)	
E. guttatus	48	48a	48	/	/	(Medrano et al., 1988)	
E. lanceolatus	48	4st + 44a	48	/	/	(Wang et al., 2003)	
E. malabaricus	48	48a	48	SCR(24)a	C(1-24)	(Zou, Yu & Zhou, 2005)	
				? (5)	EA(24)		
E. marginatus	48	48a	48	SCR(24), TR(2)	C(1-24)	(Sola et al., 2000)	
					SCR(24)a		
					TR(2)		
E. merra	48	4m + 6sm + 4st + 34a	62	/	/	(Zheng, Liu & Li, 2005)	
E. moara	48	4sm + 44a	52	SCR(24, 9)	NC(?)	Present study	
					C(?)		
					TR(?)		
E. sexfasciatus	48	2sm + 46a	50	/	/	(Chen et al., 1990)	
E. tauvina	48	2sm + 46a	50	/	/	(Raghumath & Prasad, 1980)	
Notes.

2n diploid number

a acrocentrics

C centromeric

EA nearly the entire arm

FN fundamental number

IR interstitial region

m metacentrics

NC almost indiscemible

NORs nucleolar organizer regions

SA short arm

sm submetacentrics

st subtelocentrics

SCR subcentromeric region

TR telomeric region

/ not available

? not mentioned or measured

Numbers in parentheses the number of chromosome pairs

a Data estimated from illustrations and text in the respective papers.

The cytogenetic analyses suggested that the lineages of E. bruneus and E. moara recently derived within the genus Epinephelus. Both fish species share a uniform number of chromosomes to other species in the genus Epinephelus (Wang et al., 2012). However, they contained more biarmed chromosomes such as metacentric and/or submetacentric chromosomes (Table 1). In fishes, 48 uni-armed chromosome types like acrocentric chromosomes represented the ancestral complement of diploid origin (Ohno, 1974; Vitturi et al., 1991; Sola et al., 2000). In addition, most species in genus Epinephelus showed a conserved, NOR-bearing chromosome pair No. 24 (Table 1), while E. bruneus and E. moara showed additional NORs on chromosome pairs. For most vertebrates, the presence of a single NOR pair seems to be an ancestral character state (Hsu & Pardue, 1975; Schmid, 1978; Galetti, Molina & Affonso, 2006). Both E. bruneus and E. moara show even more constitutive heterochromatin (related to chromosomal rearrangements or variation) than other species in Epinephelus (Sola et al., 2000; Molina, Maia-Lima & Affonso, 2002; Phillips & Rab, 2001; Wang et al., 2012).

Moreover, E. moara exhibited more plesiomorphic features than E. bruneus. The fundamental number (FN) of E. bruneus is larger than that of E. moara. Species with a larger FN are more derived in evolutionary terms (Martinez et al., 1989; Ghigliotti et al., 2007). Chromosomal rearrangements and genomic modifications were more obviously in E. bruneus compared to those in E. moara. Interstitial C-bandings appeared in E. bruneus imply the karyotypic rearrangement (Galetti, Aguilar & Molina, 2000), robertsonian rearrangements and/or reciprocal translocations (Eler et al., 2007). Despite the conservation of (TTAGGG)n sequence and location, slight changes in the telomeric sequences have occurred during vertebrate evolution (Meyne et al., 1990). Uniform telomeric distribution in E. moara is similar to other species in Epinephelus (Table 1) (Sola et al., 2000; Wang et al., 2012). However, remarkably high repetitions of telomere sequences seem to exist on ten chromosome pairs with stronger signals in E. bruneus, which appear to involve complex homologous or/and non-homologous recombination.

18S rDNA could be simultaneously applicable in the taxonomic and evolutionary analyses of groupers. The 5S rDNA seems to be unsuitable in the phylogenetic resolution, because the order percomorpha in the NJ trees were not recovered as monophylum. Distribution patterns of 5S rDNA of E. bruneus and E. moara are similar to other species, while that of 18S rDNA were different among species in genus Epinephelus (Sola et al., 2000; Wang et al., 2010; Wang et al., 2012). In addition, the different distributions of highly conserved18S rDNA and telomere suggest the distinct genomes and evolutionary situation of the closely related species E. bruneus and E. moara.

In summary, many useful cytogenetic charateristics are available to distinguish E. bruneus from E. moara, such as karyotypes, NORs, C-banding, 18S rDNA and telomere (TTAGGG)n distribution patterns. Moreover, the lineage of E. bruneus and E. moara seems to be derived recently, and E. moara exhibits more plesiomorphic features than E. bruneus. Molecular cytogenetic analyses could be applicable in identification of closely related species and reconstruct their phylogenetic relationships in groupers.

The authors wish to thank Miss Yan Cai, Dr. Marleen Perseke, and Dr. Li Zhang for providing language help and suggestions. We thank reviewers for their constructive comments.

Additional Information and Declarations

Competing Interests

Author Contributions

Animal Ethics

DNA Deposition

The authors declare there are no competing interests.

Guo Minglan conceived and designed the experiments, performed the experiments, analyzed the data, wrote the paper, prepared figures and/or tables.

Wang Shifeng performed the experiments, prepared figures and/or tables.

Su Yongquan analyzed the data, contributed reagents/materials/analysis tools.

Zhou Yongcan contributed reagents/materials/analysis tools, reviewed drafts of the paper.

Liu Min reviewed drafts of the paper.

Wang Jun conceived and designed the experiments.

The following information was supplied relating to ethical approvals (i.e., approving body and any reference numbers):

Institutional Animal Care and Use Committee (IACUC) of Xiamen University: 40576064.

The following information was supplied regarding the deposition of DNA sequences:

GenBank: FJ176793 and FJ176794, FJ176796 and FJ176795.

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
