# Peer review of "Molecular cytogenetic analyses of Epinephelus bruneus and Epinephelus moara (Perciformes, Epinephelidae)"

_PeerJ, doi:10.7717/peerj.412_

## Round 0.1 · original submission · Minor Revisions

In this manuscript, molecular cytogenetic analyses were used to identify the closely related species in genus Epinephelus. The reviewers both gave a recommendation of minor revisions. Please address the reviewer comments below.

·

Basic reporting

The authors report the characterization of two species of marine fish of the genus Epinephelus, whose taxonomy is still complicated and the evolutionary processes are not yet fully understood.
Emphasis should be given to multidisciplinary view of the authors to use different tools to explore this model in previous papers. In addition, cytogenetic consider as important for a better definition of the specific epithet for these cryptic species.

Experimental design

The experimental design was appropriate for the objectives proposed in this paper.

Validity of the findings

The results are clear and well representative for cytogenetic characterization species. The classical cytogenetics already points difference in the number of chromosomes of two arms between species (E. bruneus with 6 chromosomes and E. moara with 4 biarmed chromosomes).

Additional comments

The article is well presented and features important for marine fish cytogenetics, especially in relation to taxonomy, evolutionary processes and the management of fish stocks information. The techniques employed are appropriate to the purpose of the article and support the discussion. Our suggestion is that small corrections (suggested directly in the text) are carried out and that the article is accepted for publication.

Reviewer 2 ·

Basic reporting

English writing of the manuscript is not clear enough.

Experimental design

no

Validity of the findings

The data and the conclusions based on the data are acceptable.

Additional comments

There have been difficulties in identification of grouper species because of their morphological similarities. This manuscript described molecular cytogenetic analyses to identify the closely related species Epinephelus bruneus and E. moara including chromosomal karyotypes, NORs and 18S rDNA distribution patterns, heterochromatin (interstitial C-bands) and distribution pattern of telomeric sequence. The figures and tables are clearly prepared and the results provided information that E. moara is not a synonym of E. bruneus, but a distinct species. The information will be useful for our understanding of taxonomic status Epinephelidae fish. Therefore I would support the publication of this article. Nevertheless, I feel that there are many English writing and grammatical problems with the current version of the manuscript, which should be carefully corrected.

Some of the details are as follows

1. Line 24-25, Grammatical: Nucleolar organizer regions (NORs) distribution pattern and location of 18S rDNA on chromosome.
2. Line 30-31, All results confirmed that E. moara, which has long been considered avsynonym of E. bruneus, is a new species in the family Epinephelidae. Should it be "a distinct species"?
3. Line 62,what means “quantitative variation” here?
4. Line 71-72, ......techniques ......were conducted, used?
5. Line 84-86, It would be helpful to briefly describe the concentration and duration of colchicine treatment.
6. Line 90-92, "After the anatomy of fishes for the acquisition of anterior kidney, muscle tissues and fishes were stored immediately at -80℃ until used for total DNA extraction".
7. Line 96-101, "18S rDNA probes" should be 18S rDNA and 5S rDNA probes.
8. Line 137-138, "While three pairs of acrocentric chromosomes were almost indiscernible in both fish species (Fig. 4A and B)". However, it is not clear in the figure which three pairs are.
9. Line 156-157, What does the authors mean by the sentence "Two ribosomal gene families are located on different chromosome pairs"?
10. "Cytogenetic differences were inter-specific, because E. bruneus and E. moara showed a similar geographical distribution, and coupled with heterogeneously morphological characters and chromosomal structure". How "similar geo

---

## Round 0.2 · accepted · Accept

The manuscript has been well revised and can be accepted for publication.